# Using ultra-widefield red channel images to improve the detection of ischemic central retinal vein occlusion

**Akinori Sato[1], Ryo Asaoka[2,3], Shin Tanaka[1], Koichi Nagura[1], Yui Tanaka[1], Rei Arasaki[1], Kazuyoshi Okawa[1], Shohei Kitahata[1], Kentaro Nakamura[1], Shouko Ikeda[1], Tatsuya Inoue [1] *, Yasuo Yanagi[1], Maiko Maruyama-Inoue[1], Kazuaki Kadonosono[1]**

1 Department of Ophthalmology and Micro-Technology, Yokohama City University, Kanagawa, Japan,
2 Department of Ophthalmology, Seirei Hamamatsu General Hospital, Shizuoka, Hamamatsu, Japan,
3 Seirei Christopher University, Shizuoka, Hamamatsu, Japan

* inouet-tky@umin.ac.jp

**Data Availability Statement:** All relevant data are within the paper and its Supporting Information files.

## Abstract

### Purpose

To examine the usefulness of red channel fundus imaging to detect the ischemic status in eyes with central retinal vein occlusion (CRVO).

### Methods

Ultra-widefield (UWF) fundus images were obtained from 42 eyes with CRVO. Twenty-one eyes were ischemic, and 21 eyes were non-ischemic. Rubeosis was found in 11 ischemic eyes. UWF images were split into red and green channels using ImageJ software. Both the color and red channel images were used to predict the presence or absence of ischemia when examined by masked graders. The sensitivity and specificity of UWF imagings for the detection of ischemia were calculated in Group A (total 42 eyes), Group B (32 eyes excluding non-rubeotic ischemic CRVO) and Group C (31 eyes excluding rubeotic ischemic CRVO), respectively. Moreover, a linear mixed model was conducted to investigate the relationship between the type of images and the accuracy of prediction in each group.

### Results

No significant difference in the sensitivity of color fundus imaging was seen between Group A and Group B. By contrast, a significant difference in the sensitivity of red channel imaging was seen between Group A and Group B (p = 0.031). The accuracies of the predictions were not associated with the type of image in Group A and Group B, but were significantly associated in Group C (p = 0.026).

### Conclusions

UWF red channel imaging enabled more accurate detection of the ischemic status, compared with color fundus images, especially in non-rubeotic CRVO eyes.

**Funding:** The authors received no specific funding for this work.

**Competing interests:** The authors have declared that no competing interests exist.

## Introduction

Central retinal vein occlusion (CRVO) is one of the most common causes of severe vision impairment and blindness [1–3]. CRVO is classified into "ischemic" and "non-ischemic" subtypes based on the retinal capillary perfusion status. In general, ischemic CRVO is associated with a poorer visual outcome than non-ischemic CRVO [4]. Moreover, rubeotic or neovascular glaucoma is a main cause of severe vision loss in eyes with CRVO, developing in about half of ischemic CRVO cases [5]. Therefore, discriminating ischemic CRVO from non-ischemic CRVO is clinically important and is usually performed using fluorescein angiography.

Retinal hemorrhages in retinal vein occlusion (RVO) eyes are referred to as flame-shaped (splinter-shaped) and sometimes as blot or dot shaped [6], where the former is usually suggestive of hemorrhage in the superficial retinal layer and the latter is usually indicative of hemorrhage in the deep retinal layers. This difference is clinically very important, since flame-shaped hemorrhages tend to be associated with non-ischemic retinas and dot hemorrhages are related to ischemic retinas in eyes with branch retinal vein occlusion (BRVO), as revealed by a precise evaluation of the perfusion status in RVO using the recent development of optical coherence tomography angiography (OCTA) [7]. In other words, hemorrhages in the deep retinal layer are suggestive of an ischemic status in RVO patients.

Recently, the use of ultra-widefield (UWF) imaging has become widely used for the diagnosis of retinal diseases in clinical settings. UWF fundus images can be obtained in a non-contact and non-mydriatic fashion and therefore offer advantages for screening. Thus, it would be clinically advantageous if ischemic CRVO could be detected using UWF images; however, such examinations are not straightforward, particularly for non-retinal specialists such as residents. The Optos (Optos 200Tx; Optos, Dunfermline, UK) allows the visualization of the retina at 200 degrees in a single frame, and previous studies have suggested its utility for the diagnosis of several retinal diseases [8,9]. In contrast to other fundus color images, the Optos system acquires pseudocolor images by combining only red and green scanning lasers. Alhamami et al. recently reported the usefulness of splitting red channel images from color fundus images for the early detection of diabetic macular edema [10]. In general, red channel (long wavelength) images are advantageous for observing the deep retinal layer, implying that red channel imaging might be useful for the analysis of hemorrhages in the deep retinal layer when determining the ischemic status of the retina. We hypothesized that red channel imaging might be useful for evaluating the ischemic status in eyes with CRVO, even for ophthalmology residents, because retinal hemorrhages detected by red channel imaging occur in the deep retinal layer. Thus, the current study examined the usefulness of red channel images separated from UWF images to detect the ischemic status in eyes with CRVO.

## Methods

The study adhered to the principles of the Declaration of Helsinki, and written informed consent was obtained from all the eligible patients. This study was approved by the Ethics Committee of the Yokohama City University Medical Center.

### Patients

We retrospectively reviewed the medical records of consecutive patients with CRVO. All the patients underwent comprehensive ophthalmologic examinations, including a visual acuity test, slit-lamp biomicroscopy, fluorescein angiography (FA), and UWF color fundus photography using an Optos device. Visual acuity (VA) was measured in decimal units and was converted to the logarithm of the minimum angle of resolution (logMAR) units for the statistical analyses. Each UWF image was split into red and green channels using ImageJ software

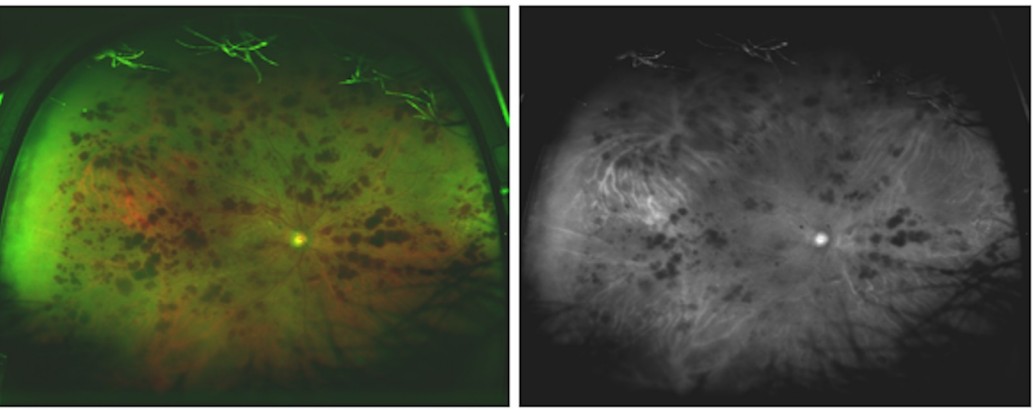

**A**                    **B**

**Fig 1. Representative image in CRVO.** Representative ultra-widefield images of ischemic CRVO. Color fundus image (A) and red channel image (B) were used for the analysis. CRVO: Central retinal vein occlusion.

(ImageJ, V.2.0.0-rc-69/1.52i; NIH, Bethesda, Maryland, USA) (Fig 1), and the usual images (red and green channels) and red channel images were used for the current analysis. The identification of "ischemic" or "non-ischemic" was performed using FA by two independent investigators who were unaware of the clinical data (A.S. and T.I.). A consensus between the investigators was reached in all the cases. Rubeotic CRVO was diagnosed when the neovascularization of the iris or angle was observed in the enrolled eyes.

## Classification by ophthalmology residents

Both image types were evaluated by 6 resident investigators (S.I., K.N., K.O., R.A., K.N. and Y. T.) who had no knowledge of any clinical data and were asked to predict the presence or absence of ischemia. These predictions were first performed using conventional images (red and green channels). Subsequently, a similar procedure was performed using the red channel images; the order of the analyzed eyes was randomized for each investigator. These predictions were then compared with the ischemic/non-ischemic diagnosis obtained using FA. The predictions were regarded as correct when they matched the true diagnosis.

## Statistical analysis

The Wilcoxon rank sum test was conducted to compare age, logMAR VA and follow-up duration between the ischemic and non-ischemic CRVO. The sensitivity and specificity for the detection of ischemic status by six resident investigators were calculated for Group A (total 42 CRVO eyes), Group B (32 eyes excluding non-rubeotic ischemic CRVO) and Group C (31 eyes excluding rubeotic ischemic CRVO), respectively. Sensitivity and specificity are usually in a balanced relation, and comparing either one alone is not recommended. Hence, a linear mixed model was conducted to investigate the relationship between the types of images (red and green channels or red channel) and the accuracy of prediction (correct or incorrect) in each of Groups A, B and C. All statistical analyses were performed using the statistical programming language R (ver. 3.4.3, The R Foundation for Statistical Computing, Vienna, Austria).

**Table 1. Subject demographics.**

|  | Total | Ischemic | Non-ischemic | P value |
|---|---|---|---|---|
| Number of eyes | 42 | 21 | 21 | - |
| Age (years) | 67.3±13.0 | 71.4±10.3 | 63.2±14.3 | 0.038 |
| Male/Female | 22/20 | 11/10 | 11/10 |  |
| LogMAR VA | 0.82±0.69 | 1.21±0.73 | 0.42±0.34 | 0.00031 |
| Follow-up duration (days) | 34.3±27.0 | 35.0±25.6 | 33.6±29.0 | 0.67 |

Data are presented as the mean ± SD.

logMAR: Logarithm of the minimum angle of resolution, VA: Visual acuity.

## Results

Forty-two eyes of 42 patients with CRVO were enrolled in total. The demographic data is shown in **Table 1**. Based on the FA results, 21 eyes were diagnosed as ischemic CRVO and 21 eyes were diagnosed as non-ischemic cases. Rubeosis was found in 11 ischemic CRVO eyes. The mean age of the participants was 67.3 years (standard deviation [SD], 13.0 years). The mean logMAR visual acuity (VA) was 0.82±0.69. A significant difference in logMAR VA was seen between the ischemic and non-ischemic groups ($p = 0.00031$, Wilcoxon rank sum test). A significant difference in age was also observed between the two groups ($p = 0.038$). The follow-up duration did not differ significantly between the two groups ($p = 0.67$).

**Table 2** shows the sensitivity and specificity of 6 resident investigators for the detection of the ischemic status in all 42 CRVO eyes (Group A), in 32 eyes excluding non-rubeotic ischemic eyes (Group B), and in 31 eyes excluding rubeotic ischemic CRVO eyes (Group C). In Group A, the sensitivity and specificity were 69.1%±16.2% and 88.9%±11.5% for the color fundus images and 71.4%±18.8% and 94.5%±7.6% for the red channel images, respectively. No significant differences in sensitivity and specificity were observed between the two imaging modalities ($p = 0.99$, $p = 0.38$, respectively, Wilcoxon rank sum test). In Group B, the sensitivity was 71.2%±18.5% for the color fundus images and 63.6%±19.9% for the red channel images. No significant difference in sensitivities was seen between the two imaging modalities in Group B ($p = 0.59$). In Group C, the sensitivity was 66.7%±16.3% for the color fundus images and 80.0%±17.9% for the red channel images, with no significant difference observed between the two imaging modalities ($p = 0.25$).

No significant difference in sensitivity was seen between Group A and Group B when the color fundus images were used ($p = 0.84$). By contrast, a significant difference in sensitivity

**Table 2. Sensitivity and specificity of two imaging modalities for detecting the ischemic status in patients with CRVO.**

| Resident investigator | Color fundus image | | | | Red channel image | | | |
|---|---|---|---|---|---|---|---|---|
|  | Sensitivity in Group A | Sensitivity in Group B | Sensitivity in Group C | Specificity | Sensitivity in Group A | Sensitivity in Group B | Sensitivity in Group C | Specificity |
| 1 | 90.5 | 100 | 80 | 90.5 | 95.2 | 90.9 | 100 | 81.0 |
| 2 | 85.7 | 81.8 | 90 | 66.7 | 81.0 | 72.7 | 90 | 90.5 |
| 3 | 47.6 | 45.5 | 50 | 95.2 | 71.4 | 63.6 | 80 | 100 |
| 4 | 61.9 | 63.6 | 60 | 90.5 | 81.0 | 72.7 | 90 | 95.2 |
| 5 | 61.9 | 72.7 | 50 | 100 | 57.1 | 45.5 | 70 | 100 |
| 6 | 66.7 | 63.6 | 70 | 90.5 | 42.9 | 36.4 | 50 | 100 |

The sensitivities of the two imaging modalities were calculated in Group A (total 42 CRVO), Group B (32 eyes: Excluding non-rubeotic ischemic CRVO) and Group C (31 eyes: Excluding rubeotic ischemic CRVO). Specificity was also calculated for the two imaging modalities.

was seen between Group A and Group B when the red channel images were used ($p = 0.031$, Wilcoxon signed rank test). A significant difference in sensitivity was also seen between Group A and Group C when the red channel images were used ($p = 0.031$); on the other hand, no significant difference was seen between the two groups when the color fundus images were used ($p = 0.84$, **Table 2**).

No significant correlation was seen between the types of images and the accuracy of predictions (correct/incorrect) in Group A ($p = 0.26$, linear mixed model). The accuracies of the predictions were not associated with the type of images in either ischemic or non-ischemic eyes ($p = 0.10$, $p = 0.67$, respectively). Similarly, a significant correlation was not observed in Group B ($p = 0.34$). On the other hand, a significant correlation was observed between the types of images and the accuracy of predictions in Group C ($p = 0.026$, linear mixed model).

## Discussion

In the present study, we investigated the usefulness of red channel images in eyes with CRVO using UWF fundus photographs. As a result, the ability of ophthalmology residents to predict the ischemic status was significantly improved when red channel images were used.

Alhamami and colleagues reported that red channel imaging provides more information about damage to deep retinal layers in patients with diabetic retinopathy [10]. Our current results were consistent with these previous findings, supporting the idea that red channel imaging is useful for detecting changes in the perfusion status of the deep retinal layer. In clinical settings, non-ischemic CRVO eyes sometimes convert to ischemic CRVO during the follow-up period. Indeed, one-third of non-ischemic CRVO cases reportedly converted to ischemic CRVO during the course of a 3-year follow-up [1], further supporting the merits of the currently proposed approach. Moreover, fluorescein angiography requires intravenous dye injection and it might lead to adverse effects, although serious allergy is actually quite rare [11,12]. Thus, the ability to predict the presence of ischemic changes non-invasively using only UWF red channel imaging would be beneficial.

OCTA is another non-invasive means of observing the ischemic status in RVO. Seknazi et al. reported a significant correlation between peripheral non-perfusion measured using FA and macular vessel density in both deep and superficial capillary plexuses measured using OCTA [13]. Also, Fukutomi et al. suggested that sequential observations using OCTA measurements were useful for detecting conversion from a non-ischemic to an ischemic status in CRVO [14]. Of note, a study using OCTA showed that macular perfusion was primarily affected in the deep retinal layer in patients with CRVO [15]. Thus, in a future study, it would be interesting to compare the usefulness of UWF red channel imaging and OCTA when evaluating the ischemic status in eyes with CRVO. However, not all real-world facilities are equipped with OCTA. Furthermore, UWF red channel imaging can be used to perform direct observations of retinal hemorrhages in the deep retinal layer not only in the macular area, but also in peripheral region, unlike OCTA.

In the current study, as shown in Table 2, the specificity in 4 of 6 resident investigators increased when using red channel images. On the other hand, one resident investigator with the best sensitivity showed a decreased specificity (Table 2). Furthermore, the sensitivity of resident investigators has an enormous range in the present study. This variation may have resulted in no significant differences in the sensitivity and specificity between two imaging modalities in Group A, B and C. On the other hand, an improvement was observed in the accuracy of diagnosing the ischemic status using red channel imaging in CRVO eyes without rubeosis (Group C), while this was not the case when all the enrolled CRVO eyes were analyzed together (Group A). In eyes with rubeosis, identification of the ischemic status may be

relatively straightforward, even using conventional color fundus images. In contrast, the merit of using red color channel image became prominent when rubeosis was not detected by the resident investigators. In other words, detecting the ischemic status in eyes without rubeosis was more challenging, and red color channel imaging was a useful approach in such cases. Supporting this hypothesis, as shown in Table 2, the sensitivity of red channel images was significantly higher in Group C than in Group A.

The current study had some limitations. Firstly, this study was retrospective and cross-sectional in nature, and the number of subjects was relatively small. It would be of a particular interest to validate the usefulness of the current approach for detecting conversion from a non-ischemic to ischemic status in longitudinal observations of the same eyes. Secondly, our current analysis using red channel imaging was qualitative, not quantitative. Clarifying the relationship between the non-perfusion area and the area of hemorrhage using red channel imaging in CRVO eyes would be interesting.

In conclusion, UWF red channel imaging enabled the accurate detection of the ischemic status, especially in non-rubeotic ischemic CRVO eyes, even when performed by residents.

## Supporting information

**S1 File. Sato CRVO final.**
(CSV)

## Acknowledgments

The authors declare that there are no financial conflicts of interest to disclose.

## Author Contributions

**Conceptualization:** Ryo Asaoka, Shin Tanaka, Tatsuya Inoue, Maiko Maruyama-Inoue, Kazuaki Kadonosono.

**Data curation:** Akinori Sato, Shin Tanaka, Koichi Nagura, Yui Tanaka, Rei Arasaki, Kazuyoshi Okawa, Shohei Kitahata, Kentaro Nakamura, Shouko Ikeda, Tatsuya Inoue.

**Investigation:** Ryo Asaoka, Rei Arasaki, Kazuyoshi Okawa, Kentaro Nakamura, Yasuo Yanagi.

**Writing – original draft:** Akinori Sato, Koichi Nagura, Tatsuya Inoue, Yasuo Yanagi, Maiko Maruyama-Inoue.

**Writing – review & editing:** Ryo Asaoka, Kazuaki Kadonosono.

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
