## [Decision Letter · Decision Letter 0]

24 Sep 2021

PONE-D-21-19313Using ultra-widefield red channel images to improve the detection of ischemic central retinal vein occlusionPLOS ONE

Dear Dr. Tatsuya Inoue,

Thank you for submitting your manuscript to PLOS ONE. After careful consideration, we feel that it has merit but does not fully meet PLOS ONE’s publication criteria as it currently stands. Therefore, we invite you to submit a revised version of the manuscript that addresses the points raised during the review process.

We look forward to receiving your revised manuscript.

Kind regards,

Alon Harris

Academic Editor

PLOS ONE

Journal Requirements:

Reviewers' comments:

Reviewer's Responses to Questions

**Comments to the Author**

1. Is the manuscript technically sound, and do the data support the conclusions?

Reviewer #1: No

Reviewer #2: Yes

2. Has the statistical analysis been performed appropriately and rigorously? 

Reviewer #1: Yes

Reviewer #2: Yes

3. Have the authors made all data underlying the findings in their manuscript fully available?

Reviewer #1: Yes

Reviewer #2: Yes

4. Is the manuscript presented in an intelligible fashion and written in standard English?

Reviewer #1: Yes

Reviewer #2: Yes

5. Review Comments to the Author

Reviewer #1: There is a great discrepancy between the results and the interpretation.

Specificity increased at 5 of the 6 reviewers, but interestingly the specificity decreased at the reviewer with the best sensitivity.

The sensitivity of the various reviewers has an enormous range from 45 to 90. The sensitivity decreased in 3 of the 6 reviewers.

All these facts were not discussed and referred to.

You can not just put everything in one pot and masking and ignore the details.

Reviewer #2: This is a well-designed and well written article. The authors do a good job of clearly explaining why using red channel imaging can be helpful to determine ischemic vs non ischemic status in eyes with CRVO (especially if no rubeosis present).

Can the authors explain in further detail why Groups B and C still included the non-ischemic eyes instead of just analyzing ischemic eyes (with and without rubeosis) separately?

It is unclear how much the findings in this paper would change management of ischemic CVRO. Do the authors think PRP or anti-VEGF therapy is warranted earlier in these ischemic CRVO patients as opposed to waiting for rubeosis ?

Page 15, line 228:

I would tone down the language a bit in this sentence, as “serious” FA induced allergy is actually quite rare (1/3800 for bronchospasm, 1/5300 for cardiovascular events, 1/220k for death). Skin reactions such as urticaria occur in 1.2% of patients but that seems to be a minor adverse event.

https://www.jacionline.org/article/S0091-6749(18)30586-4/fulltext

6. PLOS authors have the option to publish the peer review history of their article (what does this mean?). If published, this will include your full peer review and any attached files.

Reviewer #1: No

Reviewer #2: No

---

## [Author Response · Author response to Decision Letter 0]

6 Oct 2021

Academic Editor

PLOS ONE

Dear Prof. Alon Harris,

We appreciate you and the reviewers for the constructive and insightful comments. Please find our enclosed point-by-point responses (in red) to the reviewers’ comments. We hope that this revision will meet your expectations, and that our revised manuscript will better suit for publication on PLOS ONE.

Reviewer #1: There is a great discrepancy between the results and the interpretation.

Specificity increased at 5 of the 6 reviewers, but interestingly the specificity decreased at the reviewer with the best sensitivity.

The sensitivity of the various reviewers has an enormous range from 45 to 90. The sensitivity decreased in 3 of the 6 reviewers.

All these facts were not discussed and referred to. You can not just put everything in one pot and masking and ignore the details.

Thank you for the suggestion. As the Reviewer pointed out, the result of sensitivity and specificity of 6 resident investigators was not discussed in detail, therefore we have revised our manuscript as follows. 

Our result suggests that no significant differences in the sensitivity and the specificity were observed between two imaging modalities in Group A, B and C. This might be due to the data variation and we added the description in the revised manuscript.

“In the current study, as shown in Table2, the specificity in 4 of 6 resident investigators increased when using red channel images. On the other hand, one resident investigator with the best sensitivity showed a decreased specificity (Table 2). Furthermore, the sensitivity of resident investigators has an enormous range in the present study. This variation may have resulted in no significant differences in the sensitivity and specificity between two imaging modalities in Group A, B and C. On the other hand, an improvement was observed in the accuracy of diagnosing the ischemic status using red channel imaging in CRVO eyes without rubeosis (Group C), while this was not the case when all the enrolled CRVO eyes were analyzed together (Group A).”

Reviewer #2: This is a well-designed and well written article. The authors do a good job of clearly explaining why using red channel imaging can be helpful to determine ischemic vs non ischemic status in eyes with CRVO (especially if no rubeosis present).

Can the authors explain in further detail why Groups B and C still included the non-ischemic eyes instead of just analyzing ischemic eyes (with and without rubeosis) separately?

Thank you for the comment and we are sorry for the confusion. 

In the current study, we first compared the sensitivity and specificity associated with two types of images between ischemic (rubeotic + non-rubeotic) and non-ischemic CRVO eyes using Group A. However, no significant improvement in sensitivity and specificity when using red channel images was seen in Group A and no significant correlation was observed between the types of images (color or red channel image) and the accuracy of predictions. Therefore, we next divided ischemic group into rubeotic and non-rubeotic groups and statistical analyses were performed using Group B (non-ischemic + rubeotic) and Group C (non-ischemic + non-rubeotic). As a result, a significant correlation was observed between the types of images and the accuracy of predictions in Group C (p=0.026, linear mixed model). 

The reason why the improvement of prediction was seen only in Group C still remains unclear, however it is possible that the existence of rubeosis may have made the color fundus image sufficient to discriminate ischemic status in CRVO eyes. On the other hand, red channel image might be useful to detect the ischemic status in eyes without rubeosis, compared to color fundus image. 

It is unclear how much the findings in this paper would change management of ischemic CVRO. Do the authors think PRP or anti-VEGF therapy is warranted earlier in these ischemic CRVO patients as opposed to waiting for rubeosis ?

Thank you for the insightful comment. As mentioned in the Discussion, one-third of non-ischemic CRVO eyes have been reported to convert to ischemic CRVO during the 3-year’s follow-up. On the other hand, it is clinically often experienced that it is difficult to detect the ischemic eyes without a delay, because fluorescein angiography cannot be performed in all cases. The current results imply that it is clinically relevant to examine the ischemic status using the red channel image, not only the conventional colour images, so that eyes needing the fluorescein angiography is detected more accurately. . 

Page 15, line 228:

I would tone down the language a bit in this sentence, as “serious” FA induced allergy is actually quite rare (1/3800 for bronchospasm, 1/5300 for cardiovascular events, 1/220k for death). Skin reactions such as urticaria occur in 1.2% of patients but that seems to be a minor adverse event.

https://www.jacionline.org/article/S0091-6749(18)30586-4/fulltext

Thank you for the comment. According to the Reviewer’s suggestion, we have toned down the sentence.

---

## [Decision Letter · Decision Letter 1]

9 Nov 2021

Using ultra-widefield red channel images to improve the detection of ischemic central retinal vein occlusion

PONE-D-21-19313R1

Dear Dr. Tatsuya Inoue,

We’re pleased to inform you that your manuscript has been judged scientifically suitable for publication and will be formally accepted for publication once it meets all outstanding technical requirements.

Kind regards,

Alon Harris

Academic Editor

PLOS ONE

Additional Editor Comments (optional):

Reviewers' comments:

Reviewer's Responses to Questions

**Comments to the Author**

1. If the authors have adequately addressed your comments raised in a previous round of review and you feel that this manuscript is now acceptable for publication, you may indicate that here to bypass the “Comments to the Author” section, enter your conflict of interest statement in the “Confidential to Editor” section, and submit your "Accept" recommendation.

Reviewer #1: All comments have been addressed

Reviewer #2: All comments have been addressed

2. Is the manuscript technically sound, and do the data support the conclusions?

Reviewer #1: Yes

Reviewer #2: Yes

3. Has the statistical analysis been performed appropriately and rigorously? 

Reviewer #1: Yes

Reviewer #2: Yes

4. Have the authors made all data underlying the findings in their manuscript fully available?

Reviewer #1: Yes

Reviewer #2: Yes

5. Is the manuscript presented in an intelligible fashion and written in standard English?

Reviewer #1: Yes

Reviewer #2: Yes

6. Review Comments to the Author

Reviewer #1: (No Response)

Reviewer #2: The authors have sufficiently addressed my concerns from the original review. I recommend acceptance without modification.

7. PLOS authors have the option to publish the peer review history of their article (what does this mean?). If published, this will include your full peer review and any attached files.

Reviewer #1: No

Reviewer #2: **Yes: **Rehan M Hussain

---

## [Editor Report · Acceptance letter]

15 Nov 2021

PONE-D-21-19313R1 

Using ultra-widefield red channel images to improve the detection of ischemic central retinal vein occlusion 

Dear Dr. Inoue:

I'm pleased to inform you that your manuscript has been deemed suitable for publication in PLOS ONE. Congratulations! Your manuscript is now with our production department. 

Kind regards, 

on behalf of

Dr. Alon Harris 

Academic Editor

PLOS ONE